# Adverse Drug Reactions in Multimorbid Older People Exposed to Polypharmacy: Epidemiology and Prevention

**Siobhán McGettigan** [1] , **Denis Curtin** [2] **and Denis O'Mahony** [1,2,*]

1   Department of Medicine, University College Cork, T12 YN60 Cork, Ireland; s.mcgettigan@umail.ucc.ie
2   Department of Geriatric Medicine, Cork University Hospital, T12 DC4A Cork, Ireland
*   Correspondence: denis.omahony@ucc.ie

**Abstract:** Adverse drug reactions (ADRs) are frequent and represent a significant healthcare burden. ADRs are a potentially avoidable contributor to excess unscheduled hospital admissions, higher morbidity, mortality, and healthcare costs. The objective of this review is to examine the epidemiology of ADRs in older multimorbid adults and to explore strategies for ADR prevention. ADRs in this population are often linked to commonly prescribed medications, including anticoagulants, antiplatelet agents, insulin, and non-steroidal anti-inflammatory drugs, but ADRs and adverse drug events (ADEs) in fact encompass a much broader range of culprit drugs. Age-related factors such as changes in pharmacokinetics and pharmacodynamics, multimorbidity, polypharmacy, and frailty have been associated with ADR occurrences. Various strategies have been proposed to prevent ADRs in different clinical settings, such as structured routine medication review and the use of bespoke software applications to identify potentially inappropriate prescriptions and drug interactions. Although these approaches have demonstrated some improvement in the quality of prescribing, there is still a lack of consistent evidence regarding their effectiveness in preventing ADRs. The nuanced and often intricate complexities associated with older patients' pharmacotherapy necessitate a comprehensive approach to attenuate the impact of ADRs within this growing section of most populations globally.

**Keywords:** adverse drug reactions; older adults; multimorbidity; polypharmacy; frailty

## 1. Introduction

Adverse drug reactions (ADRs) refer to any harmful, unwanted, or unintended responses to a therapeutic agent, whether anticipated or not [1]. Commonplace in clinical practice, ADRs often lead to unplanned hospitalizations, especially in older adults who frequently receive multiple medications for the treatment of multiple concurrent health conditions (multimorbidity) [2]. Recognized as a health priority, ADRs are often preventable and can significantly impact health outcomes and inflate healthcare costs [3]. In 2017, the World Health Organization (WHO) Global Patient Safety Challenge: Medication Without Harm recognized medication-related harm (MRH) as a global public health issue [4]. This reflects the fact that increased life expectancy coupled with multimorbidity and polypharmacy leads to an increased incidence of MRH, especially in older adults [5].

Determining the precise prevalence of ADRs in older people is difficult due to varying ADR definitions and ADR detection methods described by investigators in the literature. O'Connor et al. [6] demonstrated a 26% prevalence of ADRs using the WHO definition and WHO-UMC causality tool, and using the same methods, O'Mahony et al. [7] reported a 21% prevalence of ADRs. Utilizing the Edwards and Aronson definition, Tangiisuran et al. [8] reported a 12.5% ADR prevalence rate. Multiple systematic reviews have demonstrated the variability amongst studies depending on the definition and detection tools used [9,10]. Many different ADR definitions, classifications, and causality tools exist, each having their own limitations. This variability makes identification and reporting of ADRs problematic,

as well as comparison between studies, particularly in older multimorbid adults with polypharmacy. Wolfe et al. demonstrated marked heterogeneity across multiple studies with the use of eight different ADR/adverse drug event (ADE) definitions [11]. Systematic reviews indicate that the median rate of hospital admissions attributable to ADRs in individuals aged $\geq$ 65 years is approximately 10% [12,13]. Consistent with this, meta-analyses indicate that one in every ten hospital admissions among older patients results directly from ADRs [14]. However, the reported prevalence rates of ADRs exhibit substantial variability, ranging from 5% to 50%, primarily due to differences in ADR definitions and identification methods [13]. The identification of ADRs in older populations poses a particular challenge, with hospital reporting systems shown to markedly underreport ADR incidence [15,16]. Globally, more than half of all ADR-related hospital admissions in older patients are considered preventable, emphasizing the importance of understanding ADR characteristics and the types of drugs involved in more commonly occurring ADRs [17]. For these reasons, the prevention of ADR-related hospital admissions necessitates the accurate identification of individuals at risk.

It is estimated that ADRs represent between the fourth and sixth most common cause of death worldwide, placing them amongst other preventable causes of death such as heart disease, cancer, and stroke [18]. A meta-analysis of 39 prospective studies, over a period of 32 years, calculated an overall 6.7% incidence of serious ADRs, and a 0.32% death rate among patients admitted to hospital because of an ADR and those experiencing an incident ADR while in hospital [19]. It is important to highlight that ADRs are significantly underreported worldwide, with estimates of up to 94% not reported by healthcare professionals [20]. ADR-related healthcare costs are highly significant, with the financial burden in Europe estimated to be EUR 79 billion in one recent report [21]. Patients who experience ADRs have a significantly higher mean difference in hospital length of stay than those patients who do not experience ADRs [22]. A contextualized assessment of the global burden of ADRs in terms of both patients, economics, and the broader public health paradigm is a current priority for the WHO [23]. In this review, the epidemiology of ADRs in older people in several settings will be discussed: community, acute hospital, and long-term (nursing home) care. The factors that make older people particularly vulnerable to ADRs will also be evaluated. Finally, interventions to prevent harm associated with ADRs will be explored.

## 2. Adverse Drug Reactions and Adverse Drug Events: Definitions and Detection Methods

### 2.1. Definition of ADRs and ADEs

An ADR is defined by the WHO as a "response to a drug which is noxious and unintended, and which occurs at doses normally used in humans for the prophylaxis, diagnosis, or therapy of disease, or for the modification of physiological function" [24]. Adverse drug events (ADEs), which have many similarities with ADRs, are defined as harm arising from a patient's exposure to a particular drug but not necessarily arising from a direct adverse effect of that drug. This definition encompasses harm caused by inappropriate use of a drug. For example, a fragility fracture occurring one week after the initiation of a hypnotic benzodiazepine in an osteoporotic 83-year-old female with no prior history of falls or fractures could be defined as an ADE, i.e., the fragility fracture is not a direct side-effect of the hypnotic but is the indirect result of the hypnotic prescription. ADRs represent a subset of ADEs whereby harm is directly caused by appropriate drug use. Another conceptual definition of an ADE is "an injury resulting from medical intervention related to a drug" [25]. This definition encompasses medication errors (ME), ADRs, overdoses, and allergic reactions [26]. A preventable ADE is an injury which is associated with a medication error. A potential ADE is a medication error which may potentially cause harm, but harm does not actually occur [27]. An ADR is causally linked to a particular drug and causes harm during normal use and within therapeutic doses of the drug. In contrast,

ADEs may include provider/prescriber error, non-adherence, or incorrect dosages [28]. The connection between MEs, ADRs, and ADEs is illustrated by Figure 1.

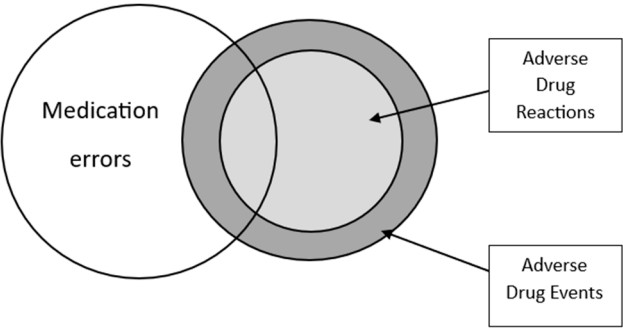

**Figure 1.** Association between MEs, ADEs, and ADRs [29].

In the published literature to date, multiple definitions of ADEs and ADRs exist (see Table 1). The definitions provided by the WHO [30] and by Laurence et al. [31] do not include administration errors, withdrawal reactions of medications, or reactions to inactive components (the definitions refer to a "drug" rather than a medicinal product). Furthermore, these definitions omit minor unwanted reactions. The definition proposed by Bates et al. [32] uses the term "injury", which introduces some ambiguity, leaving ADR assessment susceptible to subjective interpretation. Despite the distinction between ADRs and ADEs, these terms are commonly employed interchangeably, leading to challenges in reporting, interpreting, and comparing ADR incidence and prevalence in observational and interventional studies. In contrast, the ADR definition proposed by Edwards and Aronson [33] is probably the most precise, because it is clear and addresses deficiencies evident in other definitions. It also incorporates a broader range of factors, making it the most relevant definition for use in the study of ADRs in older adults. Whilst the terms ADR, ADE, and ME appear in the international literature over the last five decades, more recent updating and broadening of the definition of ADRs by the European Medicines Agency [34] has provided useful improvement in understanding what constitutes an ADR in clinical practice.

**Table 1.** Various ADE and ADR definitions from published literature.

| Authors | Definition |
|---|---|
| **Bates et al.** [32] 1995 | **ADE:** an injury resulting from medical intervention related to a drug. |
| **Edwards and Aronson** [33] 2000 | **ADR:** an appreciably harmful or unpleasant reaction, resulting from an intervention related to the use of a medicinal product, which predicts hazard from future administration and warrants prevention or specific treatment, or alteration of the dosage regimen, or withdrawal of the product. |
| **Laurence et al.** [31] 1998 | **ADR:** a harmful or significantly unpleasant effect caused by a drug at doses intended for therapeutic effect (or prophylaxis or diagnosis), which warrants a reduction in the dose or withdrawal of the drug and/or foretells hazard from future administration. |
| **Nebeker et al.** [28] 2004 | **ADE:** any physical or mental harm resulting from medication use, be it misuse, underdosing, or overdosing. |
| **WHO** [30] 1972 | **ADR:** a response to a drug that is noxious and unintended and occurs at doses normally used in man for the prophylaxis, diagnosis, or therapy of disease, or for modification of physiological function. |
| **European Medicines Agency** [34] 2010 | **ADR:** a noxious and unintended effect resulting not only from the authorized use of a medicinal product at normal doses, but also from medication errors and uses outside the terms of the marketing authorization, including the misuse and abuse of the medicinal product. |

*2.2. Classification of ADRs*

Various approaches can be utilized for the categorization of ADRs. The initial classification proposed by Thompson and Rawlins divides ADRs into Type A (i.e., exaggerated physiological responses based on known physiological effects of a drug e.g., respiratory depression with opioids) and Type B reactions (i.e., idiosyncratic unpredictable responses of a drug e.g., anaphylaxis on first exposure to penicillin) [35]. This classification was subsequently developed further by incorporating four additional reaction types [33]. Alternative

ADR classification systems include the dose, time, and susceptibility (DoTS) classification and the EIDOS classification. The DoTS method considers the drug's dosage, the timeframe of the reaction, and whether intrinsic susceptibility factors played a role in the adverse event [36]. The EIDOS classification considers five elements: the extrinsic chemical species (E) that initiates the effect; the intrinsic chemical species (I) that it affects; the distribution (D) of these chemical species in the body; the (physiological or pathological) outcome (O); and the sequela (S), which is the adverse effect [37]. These two approaches, DoTS and EIDOS, complement each other, and when used together, contribute to a more thorough definition and management of ADRs. However, their use remains largely within the research domain rather than being practical tools for day-to-day routine clinical practice.

### 2.3. ADR Causality Tools

It is often challenging for investigators to determine whether an unpleasant symptom reported by a patient is the direct result of a prescribed drug or whether there is an alternative plausible explanation. To standardize the assessment of ADRs, several causality tools have been developed. These are mainly used in the research domain. The most commonly used and cited ADR causality tools are the WHO–UMC criteria [38] and the Naranjo criteria [39]. The WHO–UMC criteria incorporate six different categories for causality: certain, probable, possible, unlikely, conditional, and unclassifiable (see Figure 2). The Naranjo causality criteria incorporate an ADR probability scale which comprises ten items to assess causality in a variety of clinical settings (see Figure 3). The WHO–UMC criteria take drug–drug interactions into consideration within its assessment, whereas the Naranjo method assesses the likelihood of one causative drug rather than the possibility of drug–drug interactions. The Naranjo method presents challenges in an older population, as it may be considered unethical to re-administer a drug (one of the Naranjo criteria) if there is significant suspicion of an ADR, and the risk of harm with re-administration is high. More recently, the Naranjo criteria have been modified to further assist in pharmacovigilance, with a moderate sensitivity (65%) and high specificity (93%) [40]. Multiple other algorithms exist, such as the Jones [41] or Karch algorithms [42], again highlighting the variation present within this domain.

| Categories | Time Sequence | Other drugs/other diseases excluded | Dechallenge effect (symptoms are better) | Rechallenge Effect (symptoms are worse) |
|---|---|---|---|---|
| Certain | Yes | Yes | Yes | Yes |
| Probable | Yes | Yes | Yes | No |
| Possible | Yes | No | No | No |
| Unlikely | No | No | No | No |

**Figure 2.** WHO–UMC Criteria for ADR causality [38].

| Question | Yes | No | Do Not Know | Score |
|---|---|---|---|---|
| 1. Are there previous conclusive reports on this reaction? | +1 | 0 | 0 | |
| 2. Did the adverse event appear after the suspected drug was administered? | +2 | −1 | 0 | |
| 3. Did the adverse reaction improve when the drug was discontinued or a specific antagonist was administered? | +1 | 0 | 0 | |
| 4. Did the adverse event reappear when the drug was re-administered? | +2 | −1 | 0 | |
| 5. Are there alternative causes (other than the drug) that could on their own have caused the reaction? Did the reaction reappear when a placebo was given? | −1 | +2 | 0 | |
| 6. Did the reaction reappear when a placebo was given? | −1 | +1 | 0 | |
| 7. Was the drug detected in blood (or other fluids) in concentrations known to be toxic? | +1 | 0 | 0 | |
| 8. Was the reaction more severe when the dose was increased or less severe when the dose was decreased? | +1 | 0 | 0 | |
| 9. Did the patient have a similar reaction to the same or similar drugs in any previous exposure? | +1 | 0 | 0 | |
| 10. Was the adverse event confirmed by any objective evidence? | +1 | 0 | 0 | |
| Total Score | | | | |

**Figure 3.** Naranjo ADR Probability Scale [43].

### 3. Epidemiology

*3.1. Why Are Older People at Increased Risk of ADRs?*

Approximately twice as many people aged ≥ 65 years are hospitalized due to ADR-related problems than their younger counterparts [44]. Similarly, among hospitalized inpatients, older age is recognized as a significant risk factor for the development of ADRs [45,46]. In the next sections, the factors that place older people at increased risk of ADRs will be discussed.

#### 3.1.1. Exposure to Polypharmacy

Polypharmacy and multimorbidity are inextricably linked, the relationship being cause-and-effect, i.e., an increasing number of diagnoses leads to an increasing number of daily medications. Multimorbidity presents a significant challenge for all physicians dealing with older people, particularly since its prevalence invariably increases with age [47]. The risk of ADRs rises with an increasing number of chronic diseases [48], and this association may be attributed to a higher likelihood of drug–disease interactions or altered drug metabolism in some older people, such as those with chronic renal and hepatic conditions [49].

The primary factor contributing to the heightened prevalence of ADRs in multimorbid older adults is polypharmacy, the by-product of multimorbidity [50]. Older individuals frequently rely on multiple medications to address various health conditions, with estimates

indicating that over 60% of the global population of adults aged over 65 years simultaneously take five or more drugs on a daily basis [50]. The potential harm arising from ADRs, drug–drug interactions and drug–disease interactions, is amplified by the increased number of prescribed medications [51]. In nursing home residents, Field et al. [52] found that the number of prescribed daily medications correlated with the risk of ADEs. Residents taking five to six daily medications had an ADR odds ratio of 2.0 (95% CI 1.2, 3.2), those taking seven to eight daily medications had an odds ratio to 2.8 (95% CI 1.7, 4.7), and those taking nine or more daily medications had an odds ratio of 3.3 (95% CI 1.9, 5.6).

### 3.1.2. Age-Related Changes in Pharmacokinetics and Pharmacodynamics

Older age, in addition to being accompanied by an increased chronic disease burden, is also associated with a range of physiological changes that alter drug pharmacokinetics (i.e., absorption, distribution, metabolism, and excretion) and pharmacodynamics (the predictability of the effect of the drug on the individual patient). Changes in body composition, blood circulation, and organ mass all contribute to altered pharmacokinetics in older adults. In healthy aging, there can be a reduction of up to 40% in hepatic blood flow and a 25–35% decrease in liver size [53]. Similar changes can be seen in renal function, with a reported 40% reduction in functioning nephrons by the eighth decade of life [54]. The inaccuracy of serum creatinine as an indicator of renal function in some older patients is a well-known age-related change due to reduced muscle mass in both sexes, more pronounced in older women [55]. These changes, which may be exaggerated by chronic disease, frailty, and declining health, can substantially influence drug metabolism and clearance, increasing the risk of ADRs due to altered bioavailability. Aging also affects sex steroid hormone levels, which can contribute to sex differences in adverse reactions to certain medications, contributing to the observation that women have a 1.5- to 1.7-fold greater risk of experiencing ADRs [56]. Altered pharmacodynamics may result in increased sensitivity to a range of drugs, including benzodiazepines, opioids, and antipsychotics, giving rise to undesirable consequences such as increased sedation, gait instability, and falls [57–59].

Frailty is another frequently encountered element of the aging process, whereby there is an age-associated decline in the reserve and function of several physiologic systems, leading to a clinically recognizable state of increased physiological vulnerability [60]. Frailty, in combination with other physiological changes, can predispose older adults to increased ADR risks. Cullinan et al. demonstrated that patients with a frailty index (FI) $\geq$ 0.16 were twice as likely to experience one or more ADRs during hospitalization and to receive a potentially inappropriate prescription as per the STOPP criteria compared to patients with a lower FI [61,62].

### *3.2. How Often Do ADRs Occur in Older People?*

In examining the incidence of ADRs in any patient population, multiple factors influence results of cohort studies including the definition and detection method of ADRs; the characteristics of the studied population; and the context of the study. The following section will explore ADR occurrences in the context of hospitalized patients, community-dwelling patients, and long-term care (LTC) patients. The most widely used ADR definitions in the literature are those of the WHO [30] and Edwards and Aronson [33]. Therefore, in this section, unless explicitly stated, it can be assumed that ADRs refer to "harmful or unpleasant reactions resulting from use of a medicinal product" and exclude administration errors, intentional and accidental poisoning, and therapeutic failure.

### 3.2.1. Hospital Setting

Hospitalized patients, by virtue of being exposed to a higher volume of newly prescribed medications, are at higher risk of developing ADRs than their community-dwelling counterparts. For this reason, and because hospitalized patients can be carefully monitored and assessed in detail for the effects of ADRs, most observational studies examining the epidemiology of ADRs have taken place in the hospital setting. Two systematic reviews

have evaluated the proportion of hospital admissions in older people that are ADR related. Oscanoa et al. [14] used a broad definition of ADRs in their inclusion criteria, encompassing reactions to overdoses, misuse, abuse, and off-label use. In addition, they included studies only involving adults aged ≥ 60 years. In contrast, Alhawassi et al. [13] limited their search to studies published between 2003 and 2013 and included only studies in adults ≥ 65 years with explicit ADR definitions and explicit methods for ascribing causality. Despite these differences, both reviews found that approximately 10% of hospital admissions in older people were partly or wholly the result of ADRs.

Jennings et al. [63] recently conducted a systematic review and meta-analysis of studies evaluating incident ADRs in hospitalized older adults. In total, 29 publications involving 27 studies and spanning 6 decades were included. While limited by marked heterogeneity in study design, patient population, ADR definition, classification, and causality tools, this review concluded that 16.9% of older adults experience one or more clinically significant ADRs while admitted to hospital. Interestingly, while diuretics were the most commonly implicated drug class in ADRs among hospitalized inpatients (19.8%), they were not an important cause of ADR-related hospital admissions in the review by Oscanoa et al. [14]. Likewise, non-steroidal anti-inflammatory drugs (NSAIDs) were a common cause of ADR-related hospital admissions but not a common cause of ADRs in hospitalized inpatients. The most commonly implicated drugs in these different clinical scenarios according to the two meta-analyses can be seen in Table 2. The differences are likely to be explained by the fact that ADRs resulting in a hospital admission are likely to be associated with significant morbidity (e.g., gastrointestinal bleeding, acute kidney injury due to NSAID use), while in a hospital setting, where patients' vital signs and blood parameters are closely monitored, relatively minor ADRs (e.g., transient hypotension, mild hypokalemia) are more likely to be detected.

**Table 2.** Drugs/drug classes most commonly implicated in ADRs in older adults.

| ADR-Related Hospital Admissions [14] | ADRs Detected in Hospitalized Patients [63] |
|---|---|
| Non-steroidal anti-inflammatory drugs | Diuretics |
| Beta-blockers | Anti-infective drugs |
| Anti-infective agents | Anti-thrombotics |
| Anti-thrombotic drugs | Drugs for obstructive airway disease |
| Digoxin | Agents acting on renin-angiotensin system |
| Agents acting on renin-angiotensin system | Psychotropics |
| Calcium antagonists | Systemic corticosteroids |
| Anti-cancer drugs | Drugs used in diabetes mellitus |
| Opioids | Cardiac medications (e.g., digoxin, |
| Drugs used in diabetes mellitus | vasodilators, anti-arrhythmics) |

### 3.2.2. Community Setting

There is just one prospective cohort study evaluating ADRs specifically in community-dwelling older adults [64]. In this study by Doherty et al., older people aged ≥ 70 years were recruited from 15 primary care practices in Ireland and followed prospectively for 6 years. ADRs were detected through review of the participants' general practice electronic medical records and linked to the national dispensed prescription medicine database and a detailed, self-reported patient postal questionnaire. In total, 592 participants were included in the study. The cumulative incidence of ADRs was 26.9% over 6 years. The majority of ADRs were mild (89.1%) and the remainder were classified as moderate. Just over one third of moderate ADRs required hospital admission. ADRs were significantly associated with female sex, polypharmacy (≥5 daily drugs), and hyperpolypharmacy (≥10 daily drugs). A strength of this study was the use of a patient self-reported questionnaire because there is evidence that between 30% and 50% of people who experience an ADR do not report their symptoms to their primary care physician [65–67]. Therefore, studies that rely on medical record review only are likely to substantially underestimate ADR incidence.

### 3.2.3. Long-Term Care/Nursing Home Setting

Several investigators have explored the occurrence of ADRs and ADEs amongst long-term care (LTC) residents. One of the more important studies by Gurwitz et al. [68] involved 1247 nursing home residents in two large academic long-term care residential facilities in Connecticut and Ontario who were observed over a prospective period of 9 months. ADEs, defined as injuries resulting from the use of particular drugs, were categorized as preventable (i.e., related to errors in prescribing, dispensing, administration, or monitoring) or non-preventable (i.e., not related to errors in these steps). Potential ADEs were reviewed initially by an 'on-the-ground' trained pharmacist before being submitted to pairs of experienced physician reviewers who independently adjudicated events using a structured implicit algorithm. Overall, there were 815 ADEs during the observation period, which equated to an incidence of 9.8 ADEs per 100 resident months. Of these, 42% were judged to be preventable. The incidence of ADEs was even higher in a very recent cohort study by Kalisch Ellett et al. [69] This prospective study followed 248 participants in 39 nursing homes in Australia and Tasmania over a period of 12 months. Participants taking four or more medications or at least one anticholinergic or sedative medication were included. Potential ADEs were identified and independently assessed by two research pharmacists to produce a short-list for further assessment by an expert clinical panel using the Naranjo Probability Scale. In total, there were 548 ADEs involving 62% of the 248 participants over the 12-month follow-up period, equating to an incidence of 20 ADEs per 100 resident months. The difference in results between the North American and Australian studies is likely explained by variation in baseline characteristics of the study participants, i.e., a relatively unselected nursing home resident population in the North American study, which contrasts somewhat with a frailer and more highly selected population in the Australian study. Differences in ADE detection methods also contributed to the study result variance.

## 4. Prevention of ADRs in Older Adults

Several investigators have evaluated interventions to reduce the risk of ADRs in older people. The most important of these interventions will be described in this section and, once again, will be discussed in relation to the setting in which they were deployed.

### 4.1. Hospital Setting

#### 4.1.1. Medication Reconciliation

Transitions of care, from community to hospital and, later, from hospital to home, are commonly associated with unintentional medication discrepancies. It is estimated that medication discrepancies complicate admission or discharge in up to 70% of patients [70–74] with nearly one third of these errors being potentially harmful [74]. Medication reconciliation, as defined by the WHO, is "the formal process in which healthcare professionals partner with patients to ensure accurate and complete medication information transfer at interfaces of care" [75]. Most commonly, this involves hospital pharmacy staff performing a comprehensive medication history at admission, detailed medication review at discharge, and sometimes post-discharge communication with primary care providers. Consistently, medication reconciliation has been shown to reduce medication discrepancies and potential adverse events, particularly when targeted at high-risk patient groups [76].

#### 4.1.2. Medication Optimization

Hospital-based interventions broadly fall into three categories: pharmacist-led interventions; physician-led interventions, and software-based interventions. Gillespie et al. [77] evaluated the impact of a clinical pharmacist intervention on drug-related morbidity in hospitalized adults aged ≥ 80 years. The intervention involved medication reconciliation and prescribing recommendations made by a senior clinical pharmacist, which were implemented at the discretion of the patients' attending physicians. In total, 400 patients were randomized to either the intervention arm or usual care arm (no clinical pharmacist input) of the study. During the study, 69% of pharmacist recommendations were implemented

by attending physicians. At 12 months, there was a statistically significant 16% reduction in all visits to the emergency department and an 80% reduction in drug-related hospital admissions in the intervention arm compared with the control arm. While promising, these results were not replicated by other investigators examining similar interventions [78,79]. O'Sullivan et al. [80] examined the impact of a software-supported structured pharmacist intervention on ADRs in hospitalized adults ≥ 65 years. The intervention involved medication reconciliation and the use of a bespoke software tool that integrated several screening tools for inappropriate prescribing. The pharmacist then provided a report document with advice points and recommendations to the attending physician for their consideration. This was a single time-point intervention. The control group received usual pharmaceutical care, which did not include routine hospital pharmacist input in all control arm patients. Overall, 737 patients were included in the study and, in the intervention arm, 55% of the pharmacist recommendations were implemented by attending physicians. ADRs occurred in 20.7% of patients in the control arm compared with 13.9% in the intervention arm (absolute risk reduction 6.8%, *p* = 0.02).

In a similar study, O'Connor et al. [81] examined the effect a physician-led intervention to reduce ADRs in a large teaching hospital in Ireland. Participants were patients aged ≥ 65 years who were admitted to hospital through the emergency department. Overall, 732 participants were included. In the intervention arm, a physician screened medications using the Screening Tool of Older Persons' Prescriptions (STOPP) and Screening Tool to Alert to Right Treatment (START) criteria to identify potentially inappropriate prescribing, and prescribing recommendations were discussed directly with the patients' attending physicians. Patients in the control arm received usual pharmaceutical care. The uptake of physician recommendations was very high at 83.4% and was associated with a statistically significant absolute risk reduction of 9.3% in ADRs associated with the intervention compared with usual care (ADRs occurred in 21% of the control arm versus 11.7% in the intervention arm). The positive results of this trial prompted the multicenter SENATOR trial [82], which evaluated the impact of software-generated STOPP/START recommendations on ADR incidence. The recommendations were presented as printed advice report documents and placed in patients' medical record for review by their primary medical team. In contrast to the trials by Gillespie et al. [77], O'Sullivan et al. [80], and O'Connor et al. [81], implementation of the software-generated advice points by attending physicians was poor at approximately 15%, and unsurprisingly, there was no significant difference in ADR incidence between the intervention and control groups. A high volume of low relevance recommendations generated by the software (giving rise to "alert fatigue") and absence of routine face-to-face interaction between primary researchers with pharmaceutical expert and attending physicians, a key component of each of the earlier trials, were among the reasons suggested for poor implementation of advice points.

Another software-based intervention was recently evaluated in the large-scale cluster randomized multicenter MedSafer trial in Canada [83], involving 5698 participants. Hospitalized adults aged ≥ 65 years who were prescribed ≥ 5 regular medications were included. Patients in the control arm received usual pharmaceutical care, involving medication reconciliation at both admission and discharge, while for patients in the treatment arm, in addition to usual pharmaceutical care, an electronic decision support tool called "Medsafer" produced deprescribing recommendations delivered to their attending physicians. Medsafer integrated data from patients' health records and arranged identified potentially inappropriate medications (PIMs) into hierarchical categories: high risk, intermediate risk, and low risk ("little value"). The reports were either filed in patients' paper-based records or displayed on their electronic health records (EHR). The primary outcome was the proportion of patients who experienced an ADR within 30 days of hospital discharge. The MedSafer trial found that the intervention did not reduce 30-day post-discharge ADR incidence compared to standard pharmaceutical care. Like the SENATOR trial, implementation of recommendations was relatively disappointing (intervention patients had 33.2% of identified PIMs deprescribed versus 17% in the control arm).

## 4.2. Community Setting

There are relatively few well-designed studies evaluating interventions to reduce ADRs in community settings. Two of the more important studies were conducted as part of the Veteran's Affairs Cooperative Study Program in the United States. Hanlon et al. [84] evaluated the impact of a multi-faceted sustained clinical pharmacist input in older people with polypharmacy attending a general medicine clinic. In total, 208 patients were randomized and followed up for 1 year. Usual care involved nurse-led review of medications and a physician visit. The intervention, in addition to usual care, involved a clinical pharmacist completing a comprehensive medication review at each clinic visit, including identification of drug-related problems, an assessment of medication appropriateness, patient education, and communication with physician stakeholders. Measures of medication appropriateness improved significantly in the intervention group over the course of the trial. While there was a 25% reduction in ADR incidence in the intervention group relative to the control group, this did not reach statistical significance, likely because the study was underpowered to detect this particular endpoint. In contrast, a statistically significant 35% reduction in the risk of serious ADRs was demonstrated by Schmader et al. [85] when recently hospitalized frail older adults were followed in specialist geriatric interdisciplinary clinics versus usual outpatient care. The duration of this trial was also 1 year, and participant numbers similar to those described by Hanlon et al. [84] were included. The larger effect size suggests that specialist interdisciplinary geriatric teams comprising physicians, nurses, and pharmacists may be better equipped to prevent, detect, and manage ADRs in older people compared to clinical pharmacists alone.

## 4.3. Long-Term Care Setting

Very few randomized clinical trials evaluating interventions to reduce ADRs in nursing home residents have been published. Gurwitz et al. [86] examined the impact of computerized provider order entry with clinical decision support on the prevention of ADRs in two large nursing home facilities. The trial involved 1118 participants who were followed for between 6 to 12 months. The trial, published more than 15 years ago, was ultimately negative, with software-related problems such as prescriber alert burden and lack of integration with existing electronic healthcare systems suggested as reasons for absence of observed efficacy.

A recent systematic review by Ali et al. [87] investigated the efficacy of pharmacist-led interventions to reduce ADRs in older residents in long-term care settings. Of note, the review included studies with methods other than randomized controlled trial design. While many of the interventions included in the review failed to reduce ADRs, the authors commented that multicomponent interventions, involving medication review, education, and collaboration with other healthcare professionals that were implemented over a longer period of time (i.e., greater than one year) were much more likely to be successful.

## 4.4. Future Direction

There are several learning points from the epidemiological studies of ADRs in older people and the clinical trials evaluating preventive strategies. First, ADRs are particularly important in the hospital setting where transitions of care (potentially resulting in prescribing errors), introduction of new medications (with increased risk of prescribing errors, drug–drug, and drug–disease interactions), and acute illness (higher risk of transiently altered pharmacokinetics and pharmacodynamics and increased risk of drug–disease interactions) render older people particularly vulnerable. Secondly, collaborative practice, where a clinical pharmacist is integrated into a team-based patient care model, is more likely to succeed in preventing harmful ADRs than a clinical pharmacist working in isolation, without access to patients' healthcare records, real-time updates in patients' clinical status, and regular face-to-face interaction with prescribers. Thirdly, software-based interventions, while likely to be important in terms of reducing prescribing errors and synthesizing large quantities of drug safety and interaction data, will enhance rather than replace the role

of clinical pharmacists in ADR detection and prevention. Finally, the most effective ADR preventive approaches require significant resources and repeated interventions.

A new concept of "polypharmacy stewardship" [76] addresses many of the shortcomings of previously described interventions and has recently been proposed as a model of care to reduce the risk of ADRs in older people (see Figure 4). It is defined as "a coordinated intervention designed to assess, monitor, improve, and measure the pharmacotherapeutic treatment of multimorbidity, taking into account potentially inappropriate medications, potential prescribing omissions, drug–drug and drug–disease interactions, and prescribing cascades, with the aim of aligning treatment regimens with the overall condition, prognosis, and preferences of the individual patient." Although the concept of polypharmacy stewardship is theoretically sound, its efficacy has not yet been tested by clinical trials compared to standard pharmaceutical care.

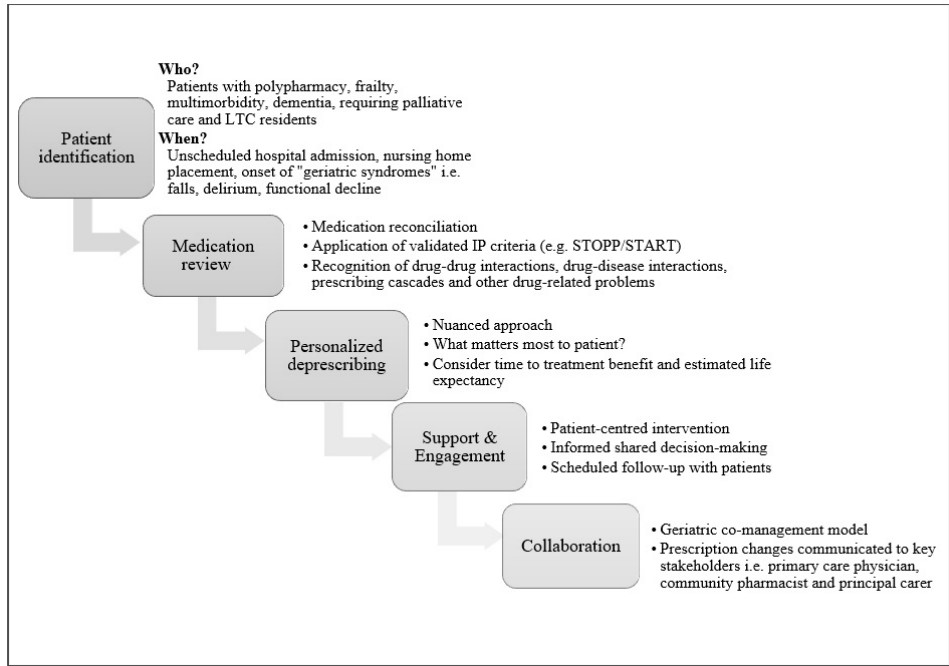

**Figure 4.** Process model for polypharmacy stewardship [88].

## 5. Conclusions

ADRs are increasingly common in older adults with multimorbidity, and associated polypharmacy and the majority are likely preventable. Approximately one in ten hospital admissions involving older adults are ADR related, and approximately one in six older adults experience clinically significant ADRs during hospital admission. These occurrences result in substantial patient harm and healthcare expenditure. For more effective ADR prevention, medication reconciliation at transitions of care and integration of clinical pharmacists and decision-support software into a team-based model of care will be required. While untested, the concept of polypharmacy stewardship holds promise as an integrated, collaborative, patient-centered approach to reducing medication-related harm.

**Author Contributions:** All authors contributed to the planning, drafting and final editing of the manuscript. All authors have read and agreed to the published version of the manuscript.

**Funding:** This research was funded by the Health Research Board of Ireland (grant number DIFA-2020-024) in the production of this manuscript.

**Conflicts of Interest:** The authors declare no conflicts of interest.

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
