# Peer review of "Adverse Drug Reactions in Multimorbid Older People Exposed to Polypharmacy: Epidemiology and Prevention"

_2813-0618, doi:10.3390/pharma3020013_

Round 1
Reviewer 1 Report
Comments and Suggestions for Authors
I read with interest the paper titled "Adverse drug reactions in multimorbid older people exposed to polypharmacy: epidemiology and prevention"
The paper is well-written. I want to adress some comments on this:
- "Determining the precise prevalence of ADRs in older people is difficult due to varying ADR definitions and ADR detection methods described by investigators in the literature." - Please provide evidence and examples of different methods, including spontaneous reporting and intensive monitoring.
- Also, it's important to highlight the remaining underreporting that exists. Global spontaneous reporting levels of ADRs are estimated to represent 5–10% of real incidence.
- Please clarify the difference between ADE and ADR. The first one could be related to the drug itself, but causality assessment was not performed.
- Please take a look of the definition of ADR by EMA regulation of 2012. ADR encompasses medication errors, abuse, misuse, off-label, among others, meaning that medication errors are now considered ADRs (after causality assessment) - Directive 2010/84/EU - please add the definition in the table and discuss consistently.
- In the sense of my previous comment, the image of ADR/ADE/ME definitions is not accurate in sense of most recent definitions, at least for Europe.
- To describe the assessment tools, you should look deeper. Recently more and more algorithms coming to discussion as recent modified Naranjo. Also, others as Jones algorithm are still used. You dont need to take all to discussion, but be in mind that the question is not limited to Naranjo and WHO binomious decision.
Epidemiology:
- In some recent studies in Europe, Netherlands used the expenditure data to correct OR to have an approximation of incidence and the potential risk associated with different ADRs, but the real prevalence remains inconclusive. This is a potential method to examine the incidence of ADRs in general population.
- Sentences like "We will now explore" should be rephrased.
- To improve readiness, I suggest swapping 3.1 with 3.2 (3.2 is important to come earlier).
- In some sentences, such as "In contrast to the trials by Gillespie et al., O’Sullivan et al., and O’Connor et al.," the references are missing.
Author Response
Dear Reviewer,
Thank you for your in-depth review, as well as your constructive comments and suggestions. They were greatly appreciated. Please find our responses below:
- "Determining the precise prevalence of ADRs in older people is difficult due to varying ADR definitions and ADR detection methods described by investigators in the literature." - Please provide evidence and examples of different methods, including spontaneous reporting and intensive monitoring. We have cited some studies here in which different ADR definitions and detection methods are used giving correspondingly variable ADR incidence results (lines 40-45).
- Also, it's important to highlight the remaining underreporting that exists. Global spontaneous reporting levels of ADRs are estimated to represent 5–10% of real incidence. We have integrated this comment into the introduction (line 69-71).
- Please clarify the difference between ADE and ADR. The first one could be related to the drug itself, but causality assessment was not performed. We have re-phrased the section title to “Adverse drug reactions and adverse drug events: definitions and detection methods”, and then re-positioned the relevant text so that the revised section runs in a logical order.
- Please take a look of the definition of ADR by EMA regulation of 2012. ADR encompasses medication errors, abuse, misuse, off-label, among others, meaning that medication errors are now considered ADRs (after causality assessment) - Directive 2010/84/EU - please add the definition in the table and discuss consistently. We have added this definition to Table 1, and mentioned it in the text (line 118-122).
- In the sense of my previous comment, the image of ADR/ADE/ME definitions is not accurate in sense of most recent definitions, at least for Europe. As above, please see line 118-122.
- To describe the assessment tools, you should look deeper. Recently more and more algorithms coming to discussion as recent modified Naranjo. Also, others as Jones algorithm are still used. You don’t need to take all to discussion, but be in mind that the question is not limited to Naranjo and WHO binomious decision. We have mentioned the modified Naranjo and other algorithms in line 186-190.
Epidemiology:
- In some recent studies in Europe, Netherlands used the expenditure data to correct OR to have an approximation of incidence and the potential risk associated with different ADRs, but the real prevalence remains inconclusive. This is a potential method to examine the incidence of ADRs in general population. Unfortunately we were unable to find the study you are referencing here, but would be happy to include it as an incidence measurement method if you wish to provide the citation.
- Sentences like "We will now explore" should be rephrased. We have rephrased these sentences (line 76-79, and line 261-262).
- To improve readiness, I suggest swapping 3.1 with 3.2 (3.2 is important to come earlier). These sections have been swapped as suggested.
- In some sentences, such as "In contrast to the trials by Gillespie et al., O’Sullivan et al., and O’Connor et al.," the references are missing. References have been added to this sentence as suggested (line 404-405).
Reviewer 2 Report
Comments and Suggestions for Authors
The authors have written a comprehensive review on ADRs and their numerous related aspects in geriatric population. While the review meticulously describes the topic, few areas of concern which I could identify are as below:
1. The topic mentions the study population as multimorbid older people however, while going through the text there are many instances where the focus in not on older population as such. For example, the initial few pages where the authors give exhaustive details on ADRs, definitions, classification and causality tools are general considerations pertaining to ADRs and very well-known facts already described in published literature. In my opinion, these areas can be cut short otherwise there is digression from the main topic making the readers lose interest in between.
Comments on the Quality of English LanguageMinor editing required.
Author Response
Dear Reviewer,
Many thanks for your valued review of our manuscript, your comments were greatly appreciated.
The topic mentions the study population as multimorbid older people however, while going through the text there are many instances where the focus in not on older population as such. For example, the initial few pages where the authors give exhaustive details on ADRs, definitions, classification and causality tools are general considerations pertaining to ADRs and very well-known facts already described in published literature. In my opinion, these areas can be cut short otherwise there is digression from the main topic making the readers lose interest in between. Thank you for this evaluation. We have removed text from within section 2 (particularly section 2.2) in an attempt to shorten the piece as suggested. However, we have kept some detail as our second reviewer recommended further additions regarding causality assessment and definitions to clarify the manuscript.
Round 2
Reviewer 2 Report
Comments and Suggestions for Authors
Thank you for making the desired changes.
Comments on the Quality of English LanguageMinor editing required.